# Essential criteria for reporting of aromatherapy-focused research in humans: An international Delphi consensus study protocol

Marian Elaine Reven[1,2]*, Esther Joy Bowles[2,3], Kelly Ablard[2,4], Marilyn Peppers-Citizen[2,5], Amanda May-Fitzgerald[2,6], Denise Joswiak[2,7], Bethany Unger[2,8]

1 School of Nursing, Health Sciences Center, West Virginia University, Morgantown, West Virginia, United States of America, 2 ARQAT | Aromatic Research Quality Appraisal Taskforce, 3 Faculty of Medicine and Health, University of New England, Armidale, New South Wales, Australia, 4 Biology Department, California State Polytechnic University, Pomona, California, United States of America, 5 Yoga Therapy Department, Maryland University of Integrative Health, Laurel, Maryland, United States of America, 6 Independent Contractor, Vista, California, United States of America, 7 Essential Health, Mendota Heights, Minnesota, United States of America, 8 The Modern Guild of Wellness, New Berlin, Wisconsin, United States of America

☯ These authors contributed equally to this work.
* marian.reven@hsc.wvu.edu

## Abstract

### Background

Reporting quality of aromatherapy-focused research in humans is inconsistent and often incomplete yet there are no (North American or American) nationally or internationally agreed upon core criteria for aromatherapy-focused research. The Aromatic Research Quality Appraisal Task Force developed the Transparent Reporting for Essential oil and Aroma Therapeutic Studies (TREATS) checklist as initial steps toward developing a reporting guideline. The purpose of this Delphi study is to engage with an international community of aromatherapy researchers to reach consensus on which items should be included in reports of aromatherapy-focused studies in humans. The result of the consensus process will be to publish an aromatherapy research reporting guideline that can be used as an extension to existing research reporting guidelines for various studies such as randomized controlled trials, observational studies, and case reports.

### Methods

A modified Delphi consensus study will be used. The consensus study, approved by the West Virginia University Institutional Review Board, will consist of up to four rounds of an online survey. To improve understanding and buy-in, experts attending a large international aromatherapy-focused conference will take part in a four-hour in-person/virtual hybrid introductory meeting where they can learn the study process and ask questions. The 48-item survey is divided into categories covering study products, processes, aromatherapy intervention, safety, sustainability, and olfactory ability and aroma preference. Participants will be asked to rate each checklist item for relevance on a 5-point Likert scale ranging from "of

**Data Availability Statement:** No datasets were generated or analysed during the current study. All

relevant data from this study will be made available upon study completion.

**Funding:** Ruth and Robert Kuhn Nursing Research Fund at West Virginia University. The funders had no role in study design, data collection and analysis, decision to publish, or preparation of the manuscript.

**Competing interests:** I have read the journal's policy and the authors of this manuscript have the following competing interests: Membership on the board of ARQAT. This does not alter our adherence to PLOS ONE policies on sharing data and materials.

little importance" to "extremely important". During the Delphi study, participants can provide comments and, in the first and second rounds, may suggest additional items or modifications to existing items. An item will be automatically included in the final guidelines if it is rated as "very important" or "extremely important" by at least $\geq$80% of the participants in Rounds 1–3, and automatically excluded if > 50% of participants rate the item as "not important" or "of little importance". Aggregated ratings will be statistically analyzed for response rates, level of agreement, medians, and interquartile ranges.

## Discussion

This protocol supports conducting a Delphi consensus that will add to the current knowledge of items considered necessary for complete and consistent reporting of aromatherapy-focused research in humans. This is of international significance as world-wide use and research of aromatherapy and essential oils in humans has continued to increase, currently without consistent and clear reporting. The Delphi method is appropriate for developing consensus between diverse experts, researchers, and practitioners as it offers anonymity and minimizes bias. Findings will contribute to creating an extension to primary reporting guidelines.

## Introduction

There is a growing interest and a large recent increase in the number of published aromatherapy-focused research articles [1, 2]. Several Cochrane systematic reviews on the use of aromatherapy for various health conditions conclude that the majority of aromatherapy research studies are either poorly designed or poorly reported and that it is difficult to draw conclusions about the effects of aromatherapy due to these limitations [3, 4]. This diminishes the potential for use in evidence-informed practice, reproducibility, summary research, policy decisions, and safety guidelines, and does not align with conventional sustainability principles.

As the research interest in essential oils and aromatherapy increases, aromatherapy researchers need to consider the ethics of using essential oils from plants that are facing extinction. At least 18% of plants from which essential oils are sourced are listed as threatened on the International Union for Conservation of Nature (IUCN) Red List [5]. Aromatherapy researchers should demonstrate consideration of environmental ethical issues surrounding essential oil choices for their research.

Many complementary therapies, including aromatherapy, require consideration of factors beyond those usually considered relevant in pharmaceutical drug trials. For example, aromatherapy-focused research should account for the effects of olfactory stimuli on mood, memory, and central nervous system stimulation, and the impact of a person's previous experience and reflexive responses to essential oils that may bias the results of an aromatherapy trial [5–7].

The Aromatic Research Quality Appraisal Taskforce (ARQAT) formed in 2021 developed and published an aromatherapy research quality appraisal checklist known as TREATS (Transparent Reporting for Essential oil and Aroma Therapeutic Studies) [8]. However, the TREATS was derived from the input of only a few dozen people, mainly from the United States of America, and it is time to consider the TREATS and additional items with a group of international aromatherapy researchers and experts to derive a consensus-based, globally created, Aromatherapy Research Reporting Guideline.

To address this need ARQAT is conducting a Delphi consensus study to determine the minimum necessary items for aromatherapy research reporting. The study participants will include English-speaking aromatherapy-focused researchers, academics, and health care and aromatherapy professionals from around the globe. The study will consist of up to four rounds of surveys delivered electronically. The first two rounds will include qualitative and quantitative data. Data will be analyzed between each round to determine consensus.

Reporting guidelines are simple, structured tools for researchers to use while writing manuscripts. They provide minimum lists of items needed to ensure that research reports can be *i*) understood by a reader, *ii*) replicated by a researcher, *iii*) used by a doctor to make a clinical decision, and *iv*) included in a systematic review [9].

## Materials and methods

### Aim

The objective is to conduct an international Delphi consensus study, to create reporting guidelines for aromatherapy-focused research in humans. These guidelines will be used in addition to existing reporting guidelines for various aromatherapy study designs including randomized controlled trials, observational studies, case reports, and other applicable studies. Existing research reporting guidelines are insufficient for prompting adequate reporting of criteria specific to aromatherapy research in humans.

### Study design and setting

This study uses a modified Delphi methodology informed by Spranger et al. (2022) whose review outlined important criteria for the conduct of Delphi studies [10]. The traditional Delphi method of having face-to-face meetings will be modified by using an online survey to enable asynchronous completion of the rounds by respondents from different countries. There will be up to four rounds (the fourth used if needed). An initial in-person/hybrid meeting will be conducted during a major aromatherapy conference, allowing potential participants to find out about the study and ask questions.

The study has been designed from a realist perspective, acknowledging that there is likely to be an optimal set of reporting criteria arising from respondents' personal beliefs and research backgrounds. The use of the modified Delphi methodology allows a robust set of items to be identified across international methods of conducting aromatherapy research. The Delphi technique has been chosen because it allows for anonymity while seeking consensus [10–12]. It is often used to determine consensus on healthcare quality indicators [13, 14].

All identified potential Delphi study participants will be sent an introductory email and Participant Information Sheet (S1 File) that will provide specific details about the project, explain consent for participation, and outline the ability to withdraw at any time. They will be invited to attend the hybrid meeting. After this meeting, all potential participants will be sent an email with the cover letter (S2 File) and a unique link to the survey. Consent to participate will be implied by the completion of the first round of the Delphi survey. Contact with participants will be via email and strategies to ensure retention will include personalized survey invitations and regular reminders. A unique identifier will be assigned to each participant.

### Participants

Participants who are proficient in reading and writing English will include representatives from two stakeholder categories: 1) researchers and academics, and 2) aromatherapy and health care professionals. People who meet these criteria will be invited to participate.

## Recruitment

Recruitment draws from the global community aiming to include as many countries as possible.

**Researchers and academics.** To recruit researchers, a literature review conducted by authors EJB and MER was used to identify authors of systematic reviews of aromatherapy-focused studies published between 2018 and 2023 (Fig 1). This identified 44 reviews whose authors documented concerns about the quality of reporting of at least one aspect of the aromatic portion of the research such as chemical constituents, Latin binomials, and quality control [8]. The authors of these reviews will be invited to join the Delphi study with the proviso that they are proficient in reading and writing English. Additional corresponding authors include those who have published aromatherapy-focused research in peer-reviewed publications available in English between 2018 and 2024 beyond the reviews noted above.

**Aromatherapy and health care professionals.** We will contact professional aromatherapy organizations and educational institutions and ask them to invite their members to express their interest. To solicit feedback from the global aromatherapy community we will draw from international organizations including the International Clinical Aromatherapy Network (ICAN) [15], the Alliance of International Aromatherapists (AIA) [16], the National Association for Holistic Aromatherapy (NAHA) [17], the International Federation of Professional Aromatherapists (IFPA) [18], International Federation of Aromatherapists (IFA) [19], the Canadian Alliance of Aromatherapy (CAOA) [20], the Canadian Federation of Aromatherapist (CFA) [21], ABRAROMA (Associação Brasileira de Aromaterapia e Aromatologia) (Brazil) [22], ARTHES (Association of Professional Aromatherapy and Care in Switzerland) (Switzerland) [23], and others.

Snowball recruitment will be used allowing participants to recommend other qualified participants for inclusion in the study. It is hoped that the global community of researchers will participate. The Participant Information Sheet, introductory email, and invitations to participate will be sent. Referrals will be accepted until the official start date of the survey.

## Sample size

In a critique of healthcare Delphi studies, Nasa et al., (2021) suggested that 30–50 participants completing the final round should be sufficient numbers for a robust consensus [24]. In a recent Delphi study involving another integrative modality, the CheckList stAndardizing the Reporting of Interventions for Yoga (CLARIFY) Delphi study [25], 25% (32) of the initial 128 participants completed the third round. Our study will aim to recruit at least 150 participants so if a similar drop-out rate is experienced, we can hope to have about 37 participants remaining in the final round.

## Criteria for the Delphi participants to consider

The items are based on the TREATS checklist with 10 extra items including two sustainability-focused items contributed by Airmid Institute.

## Item and categories

A total of 48 items have been developed and will be assessed in the Delphi rounds. These items are grouped into sections including, 1) title, 2) characterization of essential oil(s)/volatile extract(s), 3) rationale for study design and choice of plant materials, 4) aromatherapist involvement and safe handling of essential oils, 5) topical application methods and dosage regime, 6) inhalation methods and dosage regime, and 7) participant olfactory capacity and experience (Table 1).

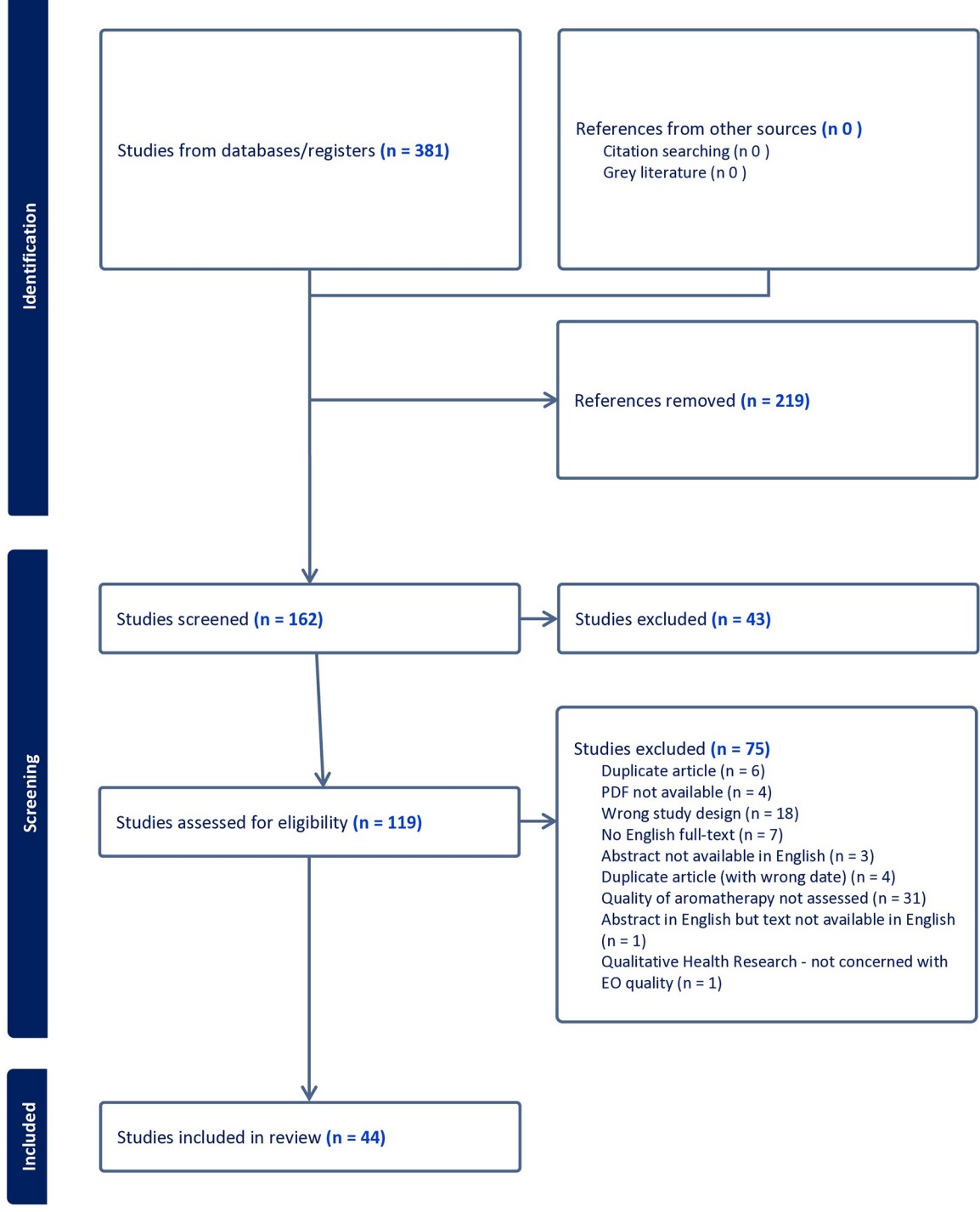

**Fig 1. PRISMA diagram of review of aromatherapy focused reviews.**

## Survey format

Up to four rounds of online surveys will be administered using the West Virginia University (WVU) REDCap platform [26, 27]. The online survey has been constructed and pilot tested for comprehension and adequate function of the survey format. Unique links for Round 1 will

**Table 1. Round one survey questions.**

| Section Header | Item |
|---|---|
| Section 1: Title | 1. Include the word "aromatherapy" in the publication title. Allows for searchability and specific content for research. |
| | 2. Include the word "essential oil" or "volatile extract" in the title. Allows for searchability and specific content for research. |
| | 3. Include the common and the botanical name of the essential oil (s)/volatile extract used for the intervention in the publication title. Allows for searchability and specific content for research. |
| Section 2: Characterization of essential oil(s)/ volatile extract(s) | 4. Include the common and full binomial (botanical, scientific) name (and chemotype if relevant) of all plant extracts used in the study. The binomial name provides identification of genus, species, and chemotype if applicable—noted by chemotype (ct) or variant (var.) |
| | 5. Include the production method (steam distillation, carbon dioxide extraction, solvent extraction, cold-pressed extraction) of the essential oil/volatile extract. The production method influences the final composition of the essential oil/volatile extract used in aromatherapy and will have different chemical components, depending on the production method. |
| | 6. Include information about the plant part used to create the essential oil/volatile extract. Some plants produce more than one type of essential oil, e.g., EOs from the Citrus aurantium plant can be obtained from the fruit peel (bitter orange), leaf/twig (petitgrain), and blossoms (neroli), all with different chemical properties |
| | 7. Include the cultivation method that provides farming method details such as pesticide/herbicide use, organic methods, and how the plant was grown and harvested. Due to varied cultivation methods and locations for growing (i.e., mountains, sea level), plants' essential/volatile oils may have varying chemical components which may impact the replication of studies. |
| | 8. Include the country of origin where the essential oil plant was grown and harvested. Essential oils/ volatile extracts produced from the same plant species grown in different countries can have different chemical composition and quality. |
| | 9. Include the name of the company that manufactured and/or sold the essential oil/volatile extract. documentation supports the determination of where the EO is manufactured and differs from where the EO is distributed. |
| | 10. Include the batch number of the essential oil/volatile extract or identify that it was extracted in the researcher's lab. The batch or lot number is a unique number provided by the supplier that will allow the essential oil/volatile extract to be traced back through its journey from plant to bottle and provide authenticity to the product. |
| | 11. Include the identification of major or complete essential oil/ volatile extract constituents listed as percentages and which analysis was performed, e.g. GC/FID, GC/MS, with Chiral Analysis, Isomeric composition, Headspace Analysis including analysis parameters, and which MS library was used for identification. Identification of plant chemical constituents in essential oils or volatile extracts is important for species identification and detection of adulteration, allowing for consistent comparison across different studies of the same oil or extract. |
| Section 3: Rationale for study design and choice of plant materials | 12. The aromatherapy definition, rationale for the definition, and the source of the definition used for the intervention are provided. There are many different operant definitions of the term "aromatherapy". The researcher describes the specific definition and source(s) used in their research. |

(*Continued*)

**Table 1.** (Continued)

| Section Header | Item |
|---|---|
| | 13. Describe the theoretical or conceptual framework as it relates to using aromatherapy in research in humans. This is important in all research; however, it is rarely included in aromatherapy research. |
| | 14. Describe the rationale for determining the selected essential oil (s) and application methods that are appropriate for the age and demographics of the participant population and that support the research study hypothesis. Examples include using specific essential oils/volatile extracts that are safe to use with known vulnerable populations. |
| | 15. The report contains a statement on whether the extract(s) used were obtained from species not classified as threatened or CITES-protected. As many extracts from threatened and CITES-protected plant species are adulterated, ethical sourcing is a necessity. Neglecting sustainability may further adulteration, have safety implications, and undermine the validity of study results-all of which are in direct opposition to our environmental accountability and the confidence that research participants have in us. |
| | 16. For extracts sourced from threatened and/or CITES-protected species, the report contains the steps for the CITES exemption process, and a summary of the measures implemented to maintain ethical sourcing standards. Not detailing CITES exemption and ethical sourcing steps can inadvertently lead to unethical procurement, bolster illegal trade, further threaten at-risk plants, raise doubts about research validity, and minimize the significance of adulteration and safety concerns. |
| Section 4: Aromatherapist involvement and safe handling of essential oils | 17. Include a clear description of, and rationale for, the outcome measures used for aromatherapy-focused research. |
| | 18. Identify that a qualified, registered, or certified aromatherapist is consulted in the design of the aromatic intervention. There are no internationally agreed-upon standards for aromatherapy educational curricula to define an aromatherapist. For the purposes of this survey, the term "qualified aromatherapist" is used to include aromatherapists with core knowledge of essential oil chemistry, safety, and application of aromatherapy with humans for physical and emotional conditions and overall well-being. |
| | 19. Describe the aromatherapy expertise, background, and training of those planning and providing the aromatic intervention. |
| | 20. Describe the safety considerations of the essential oil(s) relevant to the application method and dosage. Safety considerations differ with the various application methods and doses used with different population groups, e.g. children, pregnant women, health conditions, etc. |
| | 21. Provide information of any allergic, idiosyncratic or adverse reactions to the essential oil or control, including actions taken, or acknowledge that no adverse reactions are reported. Examples include screening and excluding participants with reported past adverse reactions or sensitivities to essential oils/volatile extracts and fragrances and reporting adverse reactions or sensitivities for participants with no known sensitivities to an essential oil/volatile extract or chemical constituent. |
| | 22. Describe how the essential oils are stored during the trial and how participants are instructed on storage. Examples include storing essential oils/ volatile extracts away from light, heat, and children. |
| Section 5: Topical application methods and dosage regime | 23. Provide the dilution of the essential oil, including the volume or weight per volume, and the name of the diluent. |
| | 24. Provide the dose of the essential oil (amount given at one time). |

(*Continued*)

**Table 1.** (Continued)

| Section Header | Item |
|---|---|
|  | 25. Describe the body part and surface area to which the essential oil is applied (e.g. hands, 10 cm). |
|  | 26. Provide the frequency of the essential oil dose. |
|  | 27. Describe the duration of the intervention in days, weeks, or months. |
|  | 28. Describe the control or placebo used in the intervention. |
|  | 29. Patient adherence with dosage regime is reported (where relevant). |
|  | 30. Provide the diluent common and binomial name. |
|  | 31. Provide the source of the diluent or carrier of the delivery system. |
| Section 6: Inhalation methods and dosage regime | 32. Describe the mode of aromatic inhalation and the delivery device used. |
|  | 33. Provide the total dose of the essential oil(s)/volatile oil(s), including the approximate distance the device is from the nose e.g. micro liters per inhalation. |
|  | 34. Provide the frequency of the essential oil dose. |
|  | 35. Provide the duration of the intervention. Example: exposed to essential oil vapor for 20 minutes. |
|  | 36. Describe the placebo or control used in the intervention including the aroma (if any) and volume used. |
|  | 37. Provide the diluent common and binomial name (if applicable), and state what odor it has, if any. |
|  | 38. State the suppliers of the diluent, carrier, and delivery system. Manufacturer or distributor of the carrier or delivery system (diffuser, patch, aromastick) used. |
| Section 7: Participant olfactory capacity and experience | 39. Participants are asked if they are experiencing anosmia (loss of smell), parosmia, hyposmia or other olfactory disorders. |
|  | 40. Participants are asked about any previous use of essential oils. |
|  | 41. Participants are asked about their preferences or aversions to any essential oils. |
|  | 42. Test environment odor control is described. Example: The intervention room is free of other odors, participants are instructed and screened to be free of scented lotions or perfumes and have no nasal congestion or allergies. |
|  | 43. Prior to the intervention, participants are asked about their expectations of the essential oil(s) used in the intervention. If the odor is recognized, it is acknowledged in the publication. Does the participant have an expectation that the essential oil or volatile extract used would have a certain effect on them e.g., using Lavandula angustifolia for sleep in the past and so the participant expects this effect. |
|  | 44. Prior to the intervention, participants are asked about their odor preference and like or dislike of the essential oil(s) aroma used in the intervention. Any biases are acknowledged under the limitations section. |
|  | 45. Participants are asked about odor recognition related to the essential oil(s) used in the intervention. If the odor is recognized, it is acknowledged in the publication. |
|  | 46. Participants are asked about their perceived aroma intensity of the essential oil(s) used in the intervention, and variations described. If the aroma is perceived as too weak or too strong, it is acknowledged in the publication. |
|  | 47. Discuss olfactory fatigue (experience losing sensitivity to odors after prolonged exposure, habituation) if applicable to the study. |

(*Continued*)

**Table 1.** (Continued)

| Section Header | Item |
|---|---|
|  | 48. Adverse effects from olfaction testing are reported as "none" or with a description of adverse effects. |
|  | 49. Comments. Please add any additional comments and be as specific as possible. If your comment refers to a specific section of the survey, please indicate it by the Section and Number. If you have references, please include them. Comments provided during Round 1 will be collected, analyzed, and evaluated for possible inclusion as new items in Round 2. |

be emailed to participants on the official starting date. We anticipate Round 1 will be the longest and will ask participants to allow for up to one hour. In Round 1, all participants are asked for demographic information. In subsequent rounds, participants are only asked for first and last names. For subsequent rounds, time required to complete the survey will be closer to 20 to 30 minutes. If a fourth round is needed, it should only take about 5 minutes. It is recommended that 4 to 6 weeks be given between rounds [10] though it may only be two weeks between rounds 2 and 3 and if needed rounds 3 and 4. Changes and updates will be communicated via email. Participants will receive up to five email reminders to complete each round before it closes.

## Rating process

Participants will be asked to rate each item on a 5-point Likert scale during each round. Relevance for inclusion means the item is believed to be necessary for minimum clear and complete reporting of aromatherapy research in humans. The Likert scale responses are:

1) of no importance,

2) of little importance,

3) important,

4) very important,

5) extremely important, and

6) not my area of expertise.

Items rated as 6) will not be included in the calculation of items to be included and excluded.

## Data handling

All data will be anonymized, aggregated, and shared with participants at the end of each Round. In Rounds 1 and 2, opportunity for comments is provided. Qualitative, free-text comments from Rounds 1 and 2 will be grouped as general comments. If any new items are suggested by participants at the end of Round 1, they will be considered for inclusion in Round 2. In Round 2, participants will be presented with items that did not reach consensus in Round 1 and asked to re-rate these remaining items. Round 2 comments will be accepted but no new items will be added to Round 3. Participants do not have to give the same score as they gave in Round 1 if their opinion has changed. If there are any items that do not reach consensus in Round 2, they will be presented again in Round 3. Round 4 will be used only if consensus remains unattained. Survey responses can be downloaded by participants for reference. An explanatory document is provided to add clarity for items in Round 1 (S3 File).

**Consensus criteria.** De-identified participant responses will be analyzed using Statistical Package for the Social Sciences (IBM SPSS) and Microsoft Excel software. Analysis will occur after each survey round and form the content of the subsequent round's survey. The *a priori* item consensus criterion for automatic inclusion in the Delphi guidelines will be a rating of 'Very important' or 'Extremely important' by ≥80% of participants, and the criterion for automatic exclusion will be a rating of 'Of no importance' or 'Of little importance' by ≥50% of participants. If less than 75% of participants have given scores of 'Important', 'Very important', or 'Extremely important', the item will also be excluded. Items not reaching inclusion or exclusion consensus, but rated as 'Important', 'Very important' or 'Extremely important' by ≥75% participants will be forwarded to the subsequent round for re-rating. As some participants may choose the "Not my area of expertise" response, the number of responses for each item will be taken into account for the calculation of the inclusion/exclusion criteria.

To summarize each round, Likert ratings for each item will be analyzed quantitatively and expressed as a percentage for each of the five Likert categories, together with 25, 50, and 75 percentile scores (median and interquartile range [IQR]).

**Termination criteria.** From Round 2, the median and IQR of any items not reaching inclusion or exclusion consensus will be compared with the median and IQR of the previous round. If these scores decrease or remain unchanged between a current and previous round, the item will be excluded; if these scores increase then the item will be forwarded to the subsequent round for re-rating.

## Data management plan

Electronic records will be handled and stored per the WVU Data Protection plan. MER is the Principal Investigator (PI) and administrator of the electronic data and the only person able to provide access to these files. Paper documents created for this project, if any, will be stored in a locked filing cabinet in a locked office at WVU. All research data will be stored per WVU Data Protection standards for at least three years and then destroyed in accordance with WVU protocols.

Amendments to this protocol will be handled by updating the protocol in the digital file and alerting research team members via email. Original protocol information will remain intact and changes tracked.

## Potential risks and risk management

Potential risks related to this study are considered minimal. Survey participants may feel inconvenienced by the process required in the study. This includes being asked to complete the required reading of the PIS, agree to participate by using the link to begin the first survey, and participate in up to three additional rounds of the survey process. To mitigate these risks, the project scope and aims will be clearly presented. The REDCap system allows participants to stop and save survey responses and return to complete as many times as needed. Additionally, we will provide a comprehensive explanatory document that participants will download and have throughout the process. This will serve to answer possible questions about the meaning of an item and help with context. Finally, participants will be instructed that they can stop participating at any time without fear of repercussions.

## Ethical considerations

Consent will be implied when a participant starts the survey, consistent with standard practice for online survey research. Ethical approval has been granted from the WVU Institutional Review Board (IRB) #2205571104.

**Table 2. Delphi survey rounds timeline.**

| Rounds | Emailed | Due back | Time & Task |
|---|---|---|---|
| 1 | October 23, 2024 | November 13, 2024 | 3 weeks survey |
| Break for holidays | | | 8 weeks analysis |
| 2 | January 8, 2025 | January 29, 2025 | 3 weeks survey |
| | | | 6 weeks analysis |
| 3 | March 12, 2025 | April 2, 2025 | 3 weeks survey |
| | | | 2 weeks analysis |
| 4 (if needed) | April 16, 2025 | May 1, 2025 | 3 weeks survey |

## Status and timeline

At the time of manuscript submission, the research will have just commenced recruitment of participants. Tentative survey rounds timeline is outlined in Table 2. All dates are subject to change and the hybrid introductory meeting is optional.

## Discussion

This study protocol focuses on the conduct of a Delphi consensus-building process to determine what are the minimum items necessary in a reporting guideline for the clear and complete reporting of aromatherapy-focused research in humans. To ensure that the resulting reporting guideline reflects a high degree of agreement among the expert panelists, a global outreach for participation has been organized.

While use of the Delphi process is on the rise across various disciplines, there is no set method by which it can or should be conducted [10]. It is suggested that the epistemological, methodological, and empirical foundation be carefully considered and reported. As closely as possible, this Delphi protocol integrates these recommendations.

Epistemological underpinnings rest on aromatherapy as a professional practice with formal education, safety, and practice standards as opposed to "do-it-yourself" therapy requiring no formal training [16–18]. Therefore, a realist perspective is applied where we assert that the Delphi technique in the chosen sample should be able to produce a relatively unbiased approximation of "true" knowledge [28].

Methodological underpinnings support sample selection, survey instrument, *a priori* rounds, feedback, evaluation, results, transparency about quality of data interpretation, discussion and limitations, and plans for dissemination. There is evidence that reporting guidelines improve the reporting quality of empirical studies [29]. This study is specifically designed to support this outcome.

The efforts of ARQAT and publication of the TREATS lay the foundation for many of the items included in Round 1 of the Delphi survey [8, 30, 31]. In addition to the 38 items in TREATS, 10 additional items were added in response to feedback and research since TREATS was published. For example, section 1 was added to allow for feedback about items to include in the title. Also, sustainability is considered with two new items in section 3. Comments are invited during Rounds 1 and 2 to allow further items to be added during Round 1.

Strengths of this study include rigorous pilot testing, use of items from the TREATS, use of best practice in Delphi studies, and aiming for a high level of global involvement. First, rigorous pilot testing of the study was completed during 2024. This included the online survey platform, the supporting documents, and the study questions. Mock Rounds 1 through 3 were completed, the data analyzed, and changes made to help ensure clear results. This included the

decision to add a scale question "Not my area of expertise" and to remove any response with this designation from the calculations [32]. This will enhance the quality of the responses, allowing those without expertise to decline answering items. Limitations include language and technology. Allowing only those who can read and write English and are able to participate in the online survey process to participate could limit responses.

## Conclusion

The aim of this study is to create an aromatherapy research reporting guideline to support clear and complete reporting of this aspect of research. The guideline will provide an extension to existing reporting guidelines. This will contribute to the growing repository of resources for researchers of integrative modalities such as aromatherapy and serve as support for editors, peer-reviewers, and policy makers to identify high-quality aromatherapy research in humans that can be used in policy and practice.

## Supporting information

**S1 File. Participant information sheet.**
(DOCX)

**S2 File. Cover letter.**
(DOCX)

**S3 File. Explanatory statement document.**
(DOCX)

**S4 File. ARQAT member list.**
(DOCX)

## Acknowledgments

The authors would like to acknowledge Amanda May-Fitzgerald, Denise Joswiak, and Bethany Unger for their significant contribution to this manuscript. We also acknowledge all past and present members of ARQAT (S4 File ARQAT Member List) including Michelle Cohen, Jerelyn Resnick, Donna Audia, Janet Tomaino, Barb Kurkas Lee, William McGilvray for their foundational contributions to this project. Additionally, thanks are extended to the WVU School of Nursing for their generous support and ongoing belief in nurse scientists as we explore the importance of comfort, caring, and the healing modality of aromatherapy in healthcare and nursing.

## Author Contributions

**Conceptualization:** Marian Elaine Reven, Esther Joy Bowles, Kelly Ablard, Marilyn Peppers-Citizen, Amanda May-Fitzgerald, Denise Joswiak, Bethany Unger.

**Data curation:** Denise Joswiak, Bethany Unger.

**Funding acquisition:** Marian Elaine Reven, Denise Joswiak.

**Investigation:** Denise Joswiak.

**Methodology:** Marian Elaine Reven, Esther Joy Bowles, Kelly Ablard, Marilyn Peppers-Citizen, Denise Joswiak, Bethany Unger.

**Project administration:** Marian Elaine Reven, Esther Joy Bowles, Denise Joswiak.

**Writing – original draft:** Marian Elaine Reven, Esther Joy Bowles, Kelly Ablard, Marilyn Peppers-Citizen, Amanda May-Fitzgerald, Denise Joswiak.

**Writing – review & editing:** Marian Elaine Reven, Esther Joy Bowles, Kelly Ablard, Marilyn Peppers-Citizen, Amanda May-Fitzgerald, Denise Joswiak, Bethany Unger.

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
