## [Decision Letter · Decision Letter 0]

20 Dec 2024

PONE-D-24-41981Essential criteria for reporting of aromatherapy-focused research in humans: An international Delphi consensus study protocolPLOS ONE

Dear Dr. Reven,

Thank you for submitting your manuscript to PLOS ONE. After careful consideration, we feel that it has merit but does not fully meet PLOS ONE’s publication criteria as it currently stands. Therefore, we invite you to submit a revised version of the manuscript that addresses the points raised during the review process.

We look forward to receiving your revised manuscript.

Kind regards,

Muhammad Junaid Farrukh

Academic Editor

PLOS ONE

Journal Requirements:

“Ruth and Robert Kuhn Nursing Research Fund at West Virginia University”

“I have read the journal's policy and the authors of this manuscript have the following competing interests: Membership on the board of ARQAT”

4. One of the noted authors is a group or consortium [ARQAT Group Amanda May-Fitzgerald, Denise Joswiak, Bethany Unger]. In addition to naming the author group, please list the individual authors and affiliations within this group in the acknowledgments section of your manuscript. Please also indicate clearly a lead author for this group along with a contact email address.

Reviewers' comments:

Reviewer's Responses to Questions

**Comments to the Author**

1. Does the manuscript provide a valid rationale for the proposed study, with clearly identified and justified research questions?

Reviewer #1: Yes

Reviewer #2: Yes

2. Is the protocol technically sound and planned in a manner that will lead to a meaningful outcome and allow testing the stated hypotheses?

Reviewer #1: Yes

Reviewer #2: Yes

3. Is the methodology feasible and described in sufficient detail to allow the work to be replicable?

Reviewer #1: Yes

Reviewer #2: Yes

4. Have the authors described where all data underlying the findings will be made available when the study is complete?

Reviewer #1: Yes

Reviewer #2: Yes

5. Is the manuscript presented in an intelligible fashion and written in standard English?

Reviewer #1: Yes

Reviewer #2: Yes

6. Review Comments to the Author

You may also provide optional suggestions and comments to authors that they might find helpful in planning their study.

Reviewer #1: This study is very interested.

I have enquiry about:

- Have you considered focusing more on the participation of experts from countries known for their abundance of aromatic plants and essential oils?

Please explain all of abbreviations in this manuscript.

Reviewer #2: • The reviewer thanks the author for their effort. The article is well written. The delivered results are interesting both from a theoretical and practical point of view. A good paper, interesting, and is worthy of being published in this journal. However, there are some minor revisions, which must first be made. The abstract should clearly indicate the relevance of the work for international research (with number and academic percentages).

1- Introduction part is very poor, the author should support the current study with previous study and present the motivation of the current study (most of references are more than 10 years old. https://doi.org/10.1016/j.mtcomm.2024.110022).

2- Improve the aim of the current study and relate this aim with the abstract.

3- Figure 4 needs more discussion in the text and the author mention all compounds.

4- Please revise all abbreviation.

5- Number all section

6- Please, the author should valorize the conclusion with more details and mention the future recommendation part to help other researchers in the future.

7- The paper has several mistakes and errors. Kindly review the manuscript again.

7. PLOS authors have the option to publish the peer review history of their article (what does this mean?). If published, this will include your full peer review and any attached files.

Reviewer #1: **Yes: **Abdulrazzaq Yahya Al-Khazzan

Reviewer #2: No

---

## [Author Response · Author response to Decision Letter 0]

10 Jan 2025

Academic Editor comments Action/Rebuttal

 Thank you for this specific guidance. The formatting for the manuscript and the title page have been updated.

MANUSCRIPT

Update include: Headings are fixed, Supporting information section moved to after references. Line 412

Names are updated. Documents have been renamed to reflect requirements. S1_file.docx

S2_file.docx, S3_file.docx. There are no legends as these are all files not tables or figures.

TITLE PAGE

Updated to reflect guidance in the PLOS title page document. As follows:

Direction about how to mention ARQAT in acknowledgements.

Consortia or other Group Authors• If there is a consortium or group author on your manuscript, please provide a note that describes where the full membership list is available for the readers.• The membership list can be listed in the Acknowledgments, in Supporting Information, or on the internet.• Consortia/Group authors can have affiliations, but it is not required.

Also, a separate document has been made for the title page per the guidelines found at https://journals.plos.org/plosone/s/file?id=ba62/PLOSOne_formatting_sample_title_authors_affiliations.pdf.

Using this formatting, the following has been added to the author byline: Amanda May-Fitzgerald, Denise Joswiak, and Bethany Unger¶

“Ruth and Robert Kuhn Nursing Research Fund at West Virginia University”

 Thank you for this correction. The following has been added to the Updated Cover Letter “Financial disclosure statement as requested: The funders had no role in study design, data collection and analysis, decision to publish, or preparation of the manuscript.”

“I have read the journal's policy and the authors of this manuscript have the following competing interests: Membership on the board of ARQAT”

Please include your updated Competing Interests statement in your cover letter; we will change the online submission form on your behalf. Thank you for this correction. The following has been added in the cover letter: Competing interests section: This does not alter our adherence to PLOS ONE policies on sharing data and materials.” (as detailed online in our guide for authors http://journals.plos.org/plosone/s/competing-interests). There are no restrictions on the sharing of data or materials.

4. One of the noted authors is a group or consortium [ARQAT Group Amanda May-Fitzgerald, Denise Joswiak, Bethany Unger]. In addition to naming the author group, please list the individual authors and affiliations within this group in the acknowledgments section of your manuscript. Please also indicate clearly a lead author for this group along with a contact email address.

 By following the provided guidelines, the authors from the ARQAT have been updated. Please see #1 and revisions within the manuscript. If this is still not correct, please provide the exact phraseology or format you wish us to have because we wish to have those three authors in the byline if at all possible. 

Reviewer #1 comments Reviewer #1 Rebuttal

 Have you considered focusing more on the participation of experts from countries known for their abundance of aromatic plants and essential oils?

 Thank you for this question. We have not focused on experts from countries that produce. Our review of the literature was focused on those countries where aromatherapy-focused studies were being published and specifically on authors of manuscripts where the quality of reporting of the aromatherapy aspects of the studies was discussed and often considered not clear and complete. 

Please explain all of abbreviations in this manuscript. 

 Please see Abbreviation_List, this is an extra document. We are unable to determine where the list is to be put in the manuscript layout. https://journals.plos.org/plosone/s/file?id=wjVg/PLOSOne_formatting_sample_main_body.pdf

Reviewer #2 comments Reviewer #2 rebuttal

The delivered results are interesting both from a theoretical and practical point of view. This is a protocol paper. There are no delivered results. We are questioning whether this Reviewer’s comments pertain to this protocol paper?

The abstract should clearly indicate the relevance of the work for international research (with number and academic percentages).

 It is unclear what “with number and academic percentages” means, or to which part of the abstract this comment relates.

1- Introduction part is very poor, the author should support the current study with previous study and present the motivation of the current study This is a protocol. There are no previous studies. We have presented the “motivation” very clearly and succinctly.

most of references are more than 10 years old. https://doi.org/10.1016/j.mtcomm.2024.110022).

 This is not true. We have only two references which are more than 10 years old, and the attached URL is irrelevant.

2- Improve the aim of the current study and relate this aim with the abstract. 

 How could the aim be made clearer?

3- Figure 4 needs more discussion in the text and the author mention all compounds. 

 The word “figure” occurs only once in our manuscript, and relates to the PRISMA diagram for inclusion of systematic reviews. We have no Figure 4, and there are certainly no “compounds” mentioned anywhere in our protocol.

4- Please revise all abbreviation. Please see Abbreviation_List, this is an extra document. 

 We are unable to determine where the list is to be put in the manuscript layout. https://journals.plos.org/plosone/s/file?id=wjVg/ PLOSOne_formatting_sample_main_body.pdf 

5- Number all section Within the manuscript direction document, there are no numbers for sections. 

 We are using the following from the editor: https://journals.plos.org/plosone/s/file?id=wjVg/PLOSOne_formatting_sample_main_body.pdf

6- Please, the author should valorize the conclusion with more details and mention the future recommendation part to help other researchers in the future.

 What does “valorize the conclusion” mean? As this is a protocol, it doesn’t make sense to include more details, as we haven’t completed the study yet. And there is no point in making future recommendations in a protocol paper.

7- The paper has several mistakes and errors. Kindly review the manuscript again. 

 This is incorrect.

---

## [Editor Report · Decision Letter 1]

15 Jan 2025

Essential criteria for reporting of aromatherapy-focused research in humans: An international Delphi consensus study protocol

PONE-D-24-41981R1

Dear Marian Reven

We’re pleased to inform you that your manuscript has been judged scientifically suitable for publication and will be formally accepted for publication once it meets all outstanding technical requirements.

Kind regards,

Muhammad Junaid Farrukh

Academic Editor

PLOS ONE
---

## [Editor Report · Acceptance letter]

20 Jan 2025

PONE-D-24-41981R1 

PLOS ONE

Dear Dr. Reven, 

I'm pleased to inform you that your manuscript has been deemed suitable for publication in PLOS ONE. Congratulations! Your manuscript is now being handed over to our production team.

Kind regards, 

on behalf of

Dr. Muhammad Junaid Farrukh 

Academic Editor

PLOS ONE